# Risk Analysis Stratification of CIN 2+ Development in a Cohort of Women Living with HIV with Negative Pap Smear and HR-HPV Test during a Long-Term Follow-Up

**DOI:** 10.3390/vaccines11020265

**Published:** 2023-01-26

**Authors:** Barbara Gardella, Alberto Agarossi, Mattia Dominoni, Isabella Pagano, Susanna Del Frati, Maria Di Giminiani, Selene Cammarata, Chiara Melito, Marianna Francesca Pasquali, Lucia Zanchi, Valeria Savasi, Arsenio Spinillo

**Affiliations:** 1Department of Clinical, Surgical, Diagnostic and Paediatric Sciences, University of Pavia, 27100 Pavia, Italy; 2Department of Obstetrics and Gynecology, IRCCS Foundation Policlinico San Matteo, 27100 Pavia, Italy; 3Department of Obstetrics and Gynecology, L. Sacco Hospital, ASST-Fatebenefratelli-Sacco, University of Milan, 20157 Milan, Italy

**Keywords:** human immunodeficiency virus, human papillomavirus, cervical intraepithelial neoplasia, colposcopy

## Abstract

Background: Women living with Human Immunodeficiency Virus are at higher risk of cervical cancer and precancer compared to women without HIV infection. The aim of the study is to evaluate the risk factors for the development of CIN2+ in a cohort of WLWH with negative colposcopy and cytology during a long follow-up period. Methods: We enrolled, in a multicentric retrospective cohort study, WLWH who attended the colposcopic services from 1999 to 2019. Patients with a normal Pap smear, a negative HR-HPV test, and at least one year of follow-up were considered for the anlysis. Results: The five-year cumulative incidence of histologically confirmed HSIL was 8.3% (95% CI = 2.6–13.6) among subjects with a CD4+ cell count of <200 cells/µL at any visit and 2.1% (95% CI = 0.7–3.4, *p* = 0.001) in women with a CD4+ cell count of persistently >200 cells/µL. In women with persistent HR-HPV infection, the five-year cumulative incidence of CIN 2+ was 6% (95% CI = 1.6–10.2) versus 2% (95% CI = 0.4–3.6, *p* = 0.012) in women without HPV infection. An HIV viremia of >200 copies/mL, a CD4+ cell count of <200 cells/µL, persistent HR-HPV infection, and smoking ≥10 cigarettes/day were all independent and statistically significant risk factors associated with the development of CIN2+ during follow-up. Conclusions: WLWH with good immune status and negative Pap smear and HR-HPV test have a low risk for CIN2+.

## 1. Introduction

Women living with the Human Immunodeficiency Virus (WLWH) are at higher risk of cervical cancer and precancer compared with women without HIV infection [1]. Globally, the literature has reported a six-fold higher risk of developing cervical cancer than the general population and that about 5% of all cervical cancers are attributable to HIV [2]. Cervical cancer mortality is about two times higher in WLWH than in women without HIV [3].

HIV infection and related immunosuppression are associated with a higher rate of acquisition, a reduced rate of clearance, and an increased persistence of Human Papillomavirus (HPV) infection [4,5]. Prolonged exposure to oncogenic HPVs increases, in these patients, the risk of developing cellular abnormalities that lead to cervical intraepithelial neoplasia (CIN) and cancer [6]. Further, low- and high-grade squamous intraepithelial lesions (LSIL and HSIL) seem to have a decreased chance of regression in WLWH compared to the general population [7,8].

While the surgical treatment of cervical cancer precursors is highly effective in immunocompetent women, with eradication in about 90% of cases, a lower efficacy has been reported in women living with HIV, in whom high rates of persistent and recurrent disease have been found in several studies [9].

The severity of immunosuppression, as indicated by a low CD4+ cell count and high HIV RNA viral load, appears to strongly affect high-risk HPV (HR-HPV) incidence and persistence and cervical lesion incidence and progression [1,10].

The impact on HPV pathogenesis of antiretroviral therapy (ART), which allows for the prolonged suppression of HIV replication and improved immune status, is still unclear, and studies evaluating this association reported conflicting results. However, increasing evidence in the literature indicates that the use of highly active antiretroviral therapy (HAART) seems to be associated with reduced HPV disease progression, especially if started at a higher CD4+ cell count and used over longer durations by adherent patients [10,11].

Several screening strategies have been proposed to prevent cervical cancer in WLWH, either with cytology alone or HR-HPV cotesting, generally characterized by shorter monitoring intervals depending on the immunological condition of the patient [12,13]. However, the optimal screening strategy for this high-risk group of women needs further assessment.

In HIV-positive women, cervical cancer screening based on HPV test was reported as positive several more fold than in the general population, and around 25–35% of women affected by HIV reported an abnormal Pap smear result (ASC-US+) during gynecological follow-up [6]. Because of this high burden of abnormal cytology, these women often undergo colposcopy and cervical biopsy, which in most cases, retrives low-grade lesions rather than CIN 2+ on histological specimens [7]. These data highlight the relevance of identifying, in this high-risk group of women, specific and positive predictive factors for cervical cancer screening in order to increase the detection of high-grade cervical precancerous lesions and decrease the number of unnecessary colposcopies and biopsies.

There are few data reported in the literature about the risk of high-grade cervical lesions in women living with HIV with a negative Pap smear and without detectable HPV infection. In this study, we evaluated the risk factors for the development of CIN2+ in a large cohort of WLWH with normal cytology and negative HPV test and normal colposcopy during a long follow-up period with the aim of identifying the most useful predictors of cervical neoplasia.

## 2. Materials and Methods

We performed a multicentric, retrospective cohort study that involved two Italian centers: IRCCS Fondazione Policlinico San Matteo, Pavia, and Luigi Sacco Hospital, ASST-Fatebenefratelli-Sacco, University of Milan, Milan. All WLWH who attended the colposcopic services of the two institutions from 1999 to 2019 were retrospectively identified by searching the clinical databases.

Those patients who presented at entry with a normal Pap smear and a negative HR-HPV test and who had at least one year of follow-up were selected for the analysis. Exclusion criteria were: age 21 years or older, ongoing pregnancy, treatment for CIN or total hysterectomy before enrollment. Institutional Review Board (IRB) approval by our institution was obtained (RC805036 IRCCS Fondazione Policlinico San Matteo, Pavia, Italy).

Anamnestic items about sociodemographic and clinical features were compiled after structured interviews during enrollment and the follow-up. All patients were evaluated every 6–12 months according to an established protocol, including a Pap smear, HPV test, and colposcopy with targeted biopsies if necessary. Abnormalities in cervical cytology were classified according to the most recent Bethesda system terminology [14]. Cytology examinations performed before the introduction of the Bethesda system terminology were revised by the pathologists of each institution involved in the study. The colposcopic examinations were recorded, taking into consideration the 2011 revised colposcopic terminology of the International Federation for Cervical Pathology and Colposcopy (IFCPC) [15]. Cervical samples for the HPV test were obtained immediately before colposcopy, and scrapes were taken with a cervix brush, suspended in Thin Prep Preserve Cyt Solution (Cytic Corporation, Marlborough, MA, USA), and stored at 4 C. HPV type-specific sequences were detected by the line probe assay INNO-LiPA HPV genotyping assay version V2 up to 2009 and version EXTRA subsequently (Fujirebio Europe, Gent, Belgium), according to the manufacturer’s instructions. The International Agency for Research on Cancer (IARC) classified HPV types into the following categories: high-risk HPV with proven carcinogenicity (16, 18, 31, 33, 45, 52, 58, 26, 35, 39, 51, 53, 56, 59, 66, 68, 73, 82) and low-risk HPVs (6, 11, 40, 43, 44, 54, 69, 70, 74) [16]. We considered persistent infection the detection of HPV DNA for at least 2 years.

The colposcopic examination was performed by two experienced gynecologists (BG and AA) certified by the Italian Society of Colposcopy and Cervico-Vaginal Pathology (SICPCV). Targeted cervical biopsies were obtained in all cases where a cervical lesion was suspected and in all cases of high-grade squamous cervical lesions (HSIL), irrespective of the colposcopic impression. Endocervical curettage was performed, according to the clinician’s judgment, when the extent of the lesion or the squamocolumnar junction was not entirely visible (transformation zone type 3). Histopathologic results were reported according to Lower Anogenital Squamous Terminology (LAST) as LSIL (CIN1), HSIL (CIN2), HSIL (CIN3), or HSIL (CIN2/3) [17]. The use and type of antiretroviral therapy were recorded on the basis of the medications listed in the patients’ charts. HAART was defined as the co-administration of two nucleoside reverse-transcriptase inhibitors and at least one of the following: a protease inhibitor, a non-nucleoside reverse-transcriptase inhibitor, or an additional nucleoside reverse-transcriptase inhibitor. All other types of ART (mono- or dual-nucleoside reverse-transcriptase inhibitors) were defined as pre-HAART. CD4+ T-lymphocyte count was considered according to the Centers for Disease Control’s revised surveillance case definition for HIV infection [18]. The classes of HIV-related immunodeficiency were reported in accordance with the WHO immunologic classification for established HIV infection [19]. These classifications identify the following classes: none or not significant for a CD4+ T-cell count of 500 cells/µL; mild for a CD4+ T-cell count of 350–499 cells/µL; advanced for a CD4+ T-cell count of 200–349 cells/µL; and severe for a CD4+ T-cell count of 200 cells/µL.

### Statistical Analysis

Univariate statistical analysis was carried out with Kruskal–Wallis analysis of variance and the chi-square test to compare continuous and categorical variables, respectively. The Bonferroni method for multiple comparisons was used to evaluate partitioned chi-square tests in multiway contingency tables. We did not estimate a preliminary sample size because this was not a randomized study but a retrospective cohort study. To evaluate the independent effects of age, CD4+ cell count, HIV viral load, HR-HPV infection, and smoking, we used a stepwise ordered logistic regression model. All the analyses were carried out with STATA 13.0 [20].

## 3. Results

Out of a total of 904 WLWH seen on a regular basis over a period of 20 years, 599 (66.3%) had a normal Pap smear, a negative HR-HPV test, and a normal colposcopy and were included in the study. They had a minimum of two years of follow-up, a median follow-up time of 84.3 months (IQR = 31–167), and a median of five visits (IQR 3–7). The main characteristics of the study population are shown in Table 1. The median age was 34 years (IQR 30–40). Most of the women (86%) were of Caucasian origin. No smoking habit was reported in 45.4% (272/599) of patients, while 22.9% (137/599) and 31.7% (190/599) smoked <10 and >10 cigarettes/day, respectively. The route of acquisition of HIV was self-reported to be by injection (drug use) in 25.6%, by sexual intercourse in 67.1%, and by other ways (perinatal transmission, blood products, or unknown) in 7.4%.

At the time of inclusion 50.75% (304/599) of women were receiving HAART, 20.37% (122/599) were receiving pre-HAART, while 28.9% (173/599) did not use any drugs; the median CD 4+ cell count was 454 cells/µL (IQR 294–600), and a low HIV viral load (<200 copies/mL) was found in 69.3% (415/599) of patients. 

During the follow-up, an incident HR-HPV infection was diagnosed in 280 patients (46.7%), and in 20.7% (124/599), the infection persisted for more than two years. The overall incidence of HR-HPV was 43.6% in women with normal cytology and increased with the severity of cervical lesions (62% in CIN1 and 70% in CIN2+). HPV-16 type has been found in 7.5% of cases, and multiple infections have been detected in 12.2% of cases; 53.3% (319/599) of women remained negative to HR-HPV testing.

The incidence of ASC-H or HSIL on a Pap smear was 9.2% (95% CI = 7–12). A histologically proven HSIL (CIN2+) was found in 30 patients (5%, 95% CI = 3.5–7.1), while LSIL (CIN1) was diagnosed in 58 women (9.7%, 95% CI = 7.5–12.5). No cases of invasive cervical cancer were detected during the study period.

Table 2 shows the univariate association between anamnestic and clinical variables and the histological diagnosis of CIN during follow-up. Compared to the negatives, women with histological CIN of any grade had significantly longer follow-up time (Spearman Rho = 0.26, *p* < 0.001), a higher number of visits (Spearman rho = 0.3, *p* < 0.001), a higher number of sexual partners (Spearman Rho = 0.11, *p* = 0.01) and a higher number of daily cigarettes smoked (Spearman Rho = 0.14, *p* = 0.001). HR-HPV persistence and multiple HR-HPV infections were significantly associated with the development of any grade of CIN, while a CD4+ cell count of <200 cells/µL and an HIV viremia of >200 copies/mL at any follow-up visit were associated with an increased incidence of CIN2+. HIV transmission routes did not play a significant role in the development of CIN. Similarly, the use and type of antiretroviral therapy were not found to be significantly associated with the risk of CIN.

The five-year cumulative incidence of histologically confirmed HSIL was 8.3% (95% CI = 2.6–13.6) among subjects with a CD4+ cell count of <200 cells/µL at any visit and 2.1% (95% CI = 0.7–3.4, *p* = 0.001) in women with a CD4+ cell count of persistently >200 cells/µL.

In women with persistent HR-HPV infection, the five-year cumulative incidence of CIN 2+ was 6% (95% CI = 1.6–10.2) versus 2% (95% CI = 0.4–3.6, *p* = 0.012) in women without HPV infection. To evaluate the independent effects of age, CD4+ cell count, HIV viral load, HR-HPV infection, and smoking, we used a stepwise ordered logistic regression model. CIN incidence was modeled as a three-level variable (no CIN, low-grade, and high-grade CIN); the age of the subjects was considered a potential confounder and was forced into the model. The results of this modeling are reported in Table 3. An HIV viremia of > 200 copies/mL, a CD4+ cell count of <200 cells/µL, persistent HR-HPV infection, and smoking ≥10 cigarettes/day were all independent and statistically significant risk factors associated with the development of CIN2+ during follow-up.

## 4. Discussion

The present study found 30 (5%) cases of CIN2+ in a cohort of 599 women living with HIV with concurrent negative cytology and an HR-HPV test at enrollment during a long-term follow-up of 20 years. Importantly, no cases of cervical cancer were diagnosed during all periods of observation. Similarly, Keller et al. [21] reported an overall five-year cumulative incidence of CIN2+ of 5% (95% CI, 2–8%) in a cohort of HIV-infected women who were cytologically normal and oncogenic HPV-negative regardless of CD4+ cell count. They found a similar risk of precancer and cancer in the control group of HIV-uninfected women with negative cotests.

Among immunocompetent women (CD4+ cell count of >200 cells/µL), we observed a five-year cumulative incidence of histological CIN2+ of 2.1% (95% CI, 0.7–3.4), significantly different from that found in more immunosuppressed women (8.3%, 95% CI, 2.6–13.6). This result is in agreement with those reported by Robbins et al. [22], who showed that in WLWH with adequate immune status and a negative cotest, the three-year risk of CIN2+ was <1%. Their data showed a low risk following a negative cotest both in HIV-positive women with a CD4+ cell count of more than 500 cells/µL and with a CD4+ cell count of 200–499 cells/µL, supporting a safe interval screening of two to three years. They found that the risk of high-grade lesions was strongly elevated in the group of women with a CD4+ cell count of fewer than 200 cells/µL. 

In our study, low CD4+ cell count (<200 cells/µL) and high HIV viremia (>200 copies/mL) at any visit, persistent HR-HPV positivity, and smoking ≥10 cigarettes/day seemed to play an independent role in the development of CIN2+.

Immunosuppression, described by a low CD4+ cell count and a high HIV viral load, is considered a well-recognized risk factor for cervical neoplasia and its precursors [1]. Several studies showed that the level of CD4+ cell count is inversely associated with the risk of HPV infection and the development of cervical disease, and women with a high CD4+ cell count (>500 cells/L) had a similar risk as those without HIV infection [10,11].

The interpretation of these observations is still debated because of the heterogeneity of CD4+ endpoints (e.g., CD4+ nadir versus current CD4+ cell count) used in the studies.

Similarly to our results, Clifford et al. [10] demonstrated that the lowest ever reported CD4+ cell count during follow-up was significantly associated with the onset of CIN2+. These findings highlight the relevance of sustained HIV viral suppression and stable high CD4+ cell count in controlling the progression of HPV infection. Some authors have reported a reduction of 36–80% of the risk of CIN2+ in women with a high nadir CD4+ cell count in comparison to those with a low nadir CD4+ cell count [10,23,24]. Konopnicki et al. [25] observed that >40 months of undetectable HIV-RNA and >18 months of CD4+ count of >500 cells/μL favor HR-HPV clearance and showed a decreased risk of persistent HR-HPV infection in patients with sustained immunological reconstitution and long-lasting HIV suppression.

The possible role of antiretroviral therapy in controlling HPV pathogenesis is still unclear. Some studies found that ART is associated with a reduction of HPV-related disease, while others found no impact of ART on cervical pathology [1]. These conflicting data may be due to several factors as the change of antiretroviral therapy regimens over time, the varying times of treatment initiation, or the poor adherence to therapy because of the high toxicity of ART drugs, in particular the older ones. The use of HAART, the most effective regimen, allowed for prolonged suppression of HIV replication and improved immune status, as manifested in a rising CD4+ cell count. It may restore cervical and vaginal mucosal immunity and seems to be associated with a reduction in the incidence and progression of cervical lesions, especially if started at a higher CD4+ cell count and used over longer durations by adherent patients [11]. In our analysis, we did not observe any independent effect of ART, both pre-HAART and HAART, on cervical disease development. This result must be interpreted with caution for several reasons. Our study started in 1999 when many patients were treated with less effective antiretroviral therapy than HAART, and often, the therapy was initiated when HIV disease was more advanced. Moreover, the regimens of antiretroviral therapy varied over time and were often self-reported by the patients, resulting in possible imprecision of classification. Further, many patients discontinued or took the therapy intermittently for drug toxicity. On the other hand, the impact of ART on HPV pathogenesis is mediated by immune system recovery and virological suppression, which makes it difficult to evaluate the true direct effect of ART.

The overall HR-HPV cumulative incidence of 47% found in our study is consistent with the data literature. Heard et al. [26] observed a prevalence rate of 49% in 518 European women infected with HIV. Clifford et al. [27] reported in their meta-analysis of 19,883 WLWH from 86 studies worldwide that the overall HR-HPV prevalence was 57%. Among WLWH with normal cytological findings, HPV prevalence was 41%, and the prevalence appeared to vary widely by region, ranging from 25–34% in Asia, Europe, and North America, up to 57–64% in Africa and Latin America. As expected, the prevalence increased with the severity of cervical disease up to 91% in invasive cervical cancer. Similarly to these data, we found an HR-HPV prevalence of 44% in women with normal Pap smears, which increased to 60% and 70% in histologically CIN1 and CIN2+ cases.

We found only 6% of women reported HPV16 incident infection and 1.6% HPV16 persistent infection; 20.7% of women showed persistent HR-HPV infection during follow-up.

In our study, the prolonged duration (more than two years) of oncogenic HPV infection appeared strongly associated with the risk of CIN2+ with an odds ratio of 3.43 (95% CI 1.97–6.01) compared to women without HPV infection or with transient one. As in the general population, the persistence of oncogenic HPV infection is the main factor for the development of cervical neoplasia. Several studies showed that WLWH with HPV infection lasting longer than six months had a higher incidence of CIN than those with transient infection [28,29,30]. HIV-induced immune suppression enhances the risk of HPV-associated tumors by being permissive for chronic HPV infection, thus allowing premalignant lesions to accumulate genetic damage and progress through increasing dysplasia to cancer. Our results support the use of the HPV test as a screening tool for this group of women at a high risk for cervical neoplasia to identify those women with a lower risk of high-grade lesions who could undergo less intensive follow-up.

Another significant result of our study is the role of cigarette smoking as an independent risk factor for CIN2+ after correcting for other variables. In our cohort of women living with HIV, 55% smoked, and about 60% of them smoked more than 10 cigarettes per day. Smoking is a modifiable risk factor for cancer. Patients at risk for HIV have relatively high rates of smoking compared to the general population, and over 70–85% of some HIV-infected populations smoke [31]. Smoking increases the risk of several important cancers affecting people with HIV, including lung, head and neck, esophageal, stomach, pancreatic, liver, and anal cancers. Smoking has been shown to be associated with cervical cancer and its precursors with a dose–response relationship. Cigarette smoke carcinogens present in the cervical mucus could interact with oncogenes and negatively affect immune function [32,33]. In particular, NK activities in smokers are significantly reduced in comparison to those of non-smokers, and smoking seems to promote the acquisition or persistence of HPV infection through a reduction in the number of CD4+ lymphocytes, Langerhans cells, and local immune responses in the cervix [32,33,34]. These data are confirmed by a recent review and meta-analysis that showed a positive association between smoking and cervical cancer, independently from HPV infection, indicating that this association is dose-related and declines after the interruption of smoking [35]. In WLWH, the destruction of CD4+ cells by HIV combined with the effect of smoke may increase the likelihood of persistent HPV and the development of HPV-related lesions. This observation implies that smoke cessation should be part of the virtuous behavior in terms of prevention to propose to all HIV-infected women.

The major strengths of our study are the large cohort of cytologically and virologically negative WLWH considered for the analysis, the extensive look at clinical data, and the long-term follow-up (a median of 7 and up to 20 years). Furthermore, we used restrictive criteria to define both the outcome (histologically HSIL+/CIN2+) and the status of HPV infection (defined as persistent when lasting longer than two years).

The limitations of this study are its retrospective nature and the possible selection bias due to the enrollment of patients from two referral centers for cervical pathology and sexually transmitted diseases. The women included in the study were followed by a specific program of screening that was not universally adopted. Some patients did not participate in all planned visits.

Moreover, HR-HPV testing and genotyping were not performed in all the included women because they only entered clinical practice in 2002. In addition, generalizations from these data may be limited because of the heterogeneous immunologic management of the women enrolled in the study and because of the temporal changes in antiretroviral therapy use over a very long follow-up period.

## 5. Conclusions

In conclusion, in HIV-positive women with negative HR-HPV test and normal Pap smear, the maintenance of an efficient and stable immune status, as indicated by a high CD4+ cell count and suppressed HIV viral load, appeared to be the main factor predicting a low risk for high-grade cervical lesions and cancer. Providing these conditions are met, WLWH could be screened in the same way as the general population. A decrease in immunological function, such as low CD4+ cell count and increased HIV viral load, should prompt the intensification of the screening. All women living with HIV should be counseled to abandon the habit of smoking.

## Figures and Tables

**Table 1 vaccines-11-00265-t001:** The main characteristics of the population considered in the study at entry.

	Number of Patients = 599
Age (years)	34 (30–40)
Ethnic origins Caucasian African Others	516 (86.1) 64 (10.7) 19 (3.2)
Cigarette smoking None <10/day ≥10/day	272 (45.4) 137 (22.9) 190 (31.7)
Number of lifetime partners	4 (2–10)
Risk factor for HIV infection Sexual contact Injection drug use Other	402 (67.1) 153 (25.6) 44 (7.4)
Antiretroviral drugs at entry None Pre-HAART HAART	173 (28.9) 122 (20.4) 304 (50.7)
Number of CD4+ cells/µL at entry	454 (294–600)
HIV VL > 200 copies/mL at entry No Yes	415 (69.3) 184 (30.7)
Months of follow-up	84.3 (31–167)
Number of visits	5 (3–7)
HR-HPV infection during FU No Incident Persistent Overall	319 (53.3) 156 (26.0) 124 (20.7) 280 (46.7%)
HPV-16 infection During FU No Incident Persistent Overall	554 (92.5) 36 (6.0) 9 (1.5) 45 (7.5%)
Multiple HR-HPV during FU	73 (12.2)
ASC-H or HSIL during FU	55 (9.2)
CIN during FU No Low grade High grade	511 (85.3) 58 (9.7) 30 (5.0)

Legend: HAART: Highly Active Antiretroviral Therapy; HR-HPV: high-risk Human Papillomavirus; ASC-H: Atypical squamous cells cannot exclude high-grade squamous intraepithelial lesion; HSIL: high-grade squamous intraepithelial lesion. FU: follow up. Results are reported as median (IQR) or N (%).

**Table 2 vaccines-11-00265-t002:** Univariate association between anamnestic and clinical variables and subsequent cervical histological diagnoses during follow-up.

	Negative *n* = 511	CIN1 *n* = 58	CIN 2+ *n* = 30
Age	34 (30–41)	32 (28–38)	34 (31–39)
Ethnic origin Caucasian African Others	437 (85.59) 59 (11.59) 15 (2.94)	51 (87.93) 3 (5.17) 4 (6.9)	28 (93.33) 2 (6.67) 0 (-)
Cigarette smoking No <10/day ≥10/day	244 (47.75) 114 (22.31) 153 (29.94)	20 (34.48) 13 (22.41) 25 (43.1)	8 (26.67) 10 (33.33) 12 (40)
Number of lifetime partners	4 (2–8)	7 (3–15)	5.5 (2–10)
Risk factors for HIV Sexual contact Injection drug use Other	348 (68.1) 123 (24.1) 40 (7.8)	34 (58.6) 21 (36.2) 3 (5.2)	20 (66.7) 9 (30.0) 1 (3.3)
CD4+ cells/µL at entry	477 (311–639)	391 (194–500)	260 (78–500)
Months of follow-up	86 (32–144)	164 (109–233)	104 (59–216)
Number of visits	5 (2–7)	10 (6–13)	6 (3–14)
HR-HPV infection No	288 (56.4)	22 (37.9)	9 (30.0)
Incident	138 (27.0)	12 (20.7)	6 (20.0)
Persistent	85 (16.6)	24 (41.4)	15 (50.0)
Overall	223 (43.6)	36 (62.1)	21 (70)
HPV-16 infection No Incident Persistent Overall	476 (93.1) 29 (5.7) 6 (1.2) 35 (6.84)	49 (84.5) 7 (12.1) 2 (3.4) 9 (15.51)	29 (96.7) 0 (0.0) 1 (3.3) 1 (3.3)
Multiple HR-HPV infection	51 (10.0)	14 (24.1)	8 (26.7)
CD4+ <200 cells/µL (any visit) No Yes	435 (85.1) 76 (14.9)	39 (67.2) 19 (32.8)	16 (53.3) 14 (46.7)
HIV VL > 200 copies/mL (any visit) No Yes	374 (73.2) 137 (26.8)	28 (48.3) 30 (51.7)	13 (43.3) 17 (56.7)

Legend: HR-HPV: high-risk Human Papillomavirus. Results are reported as median (IQR) or N (%).

**Table 3 vaccines-11-00265-t003:** Odds ratio (OR) and 95% confidence interval (CI) of CIN2+ development associated with age, CD4+ cell count, HIV viral copies, HR-HPV infection, and cigarette smoking.

	Odds Ratio	95% CI	*p*
Age (years)	0.98	0.94–1.01	0.175
HIV viral load > 200 copies/mL (any visit)			
no	ref	ref	ref
yes	2.22	1.36–3.63	0.001
HR-HPV infection			
no	ref	ref	ref
Incident	1.14	0.59–2.14	0.679
Persistent	3.43	1.97–6.01	<0.001
CD4+ < 200 cells/µL (any visit)			
no	ref	ref	ref
yes	2.77	1.64–4.63	<0.001
Cigarette smoking			
no	ref	ref	ref
<10 cigarettes/day	1.79	0.94–3.37	0.071
≥10 cigarettes/day	1.89	1.082–3.34	0.026

Legend: HR- HPV: high-risk Human Papillomavirus.

## Data Availability

The datasets generated and analyzed during the current study are available from the corresponding author upon reasonable request. Informed consent was obtained from all subjects involved in the study.

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
