# Peer review of "Risk Analysis Stratification of CIN 2+ Development in a Cohort of Women Living with HIV with Negative Pap Smear and HR-HPV Test during a Long-Term Follow-Up"

_vaccines, 2023, doi:10.3390/vaccines11020265_

Round 1

Reviewer 1 Report

This is a retrospective study evaluating the risk factors for the development of CIN2+ in a cohort of women living with HIV with negative colposcopy and cytology during a long follow up (1999-2019).

This is a well written study concerning an important problem, six-fold higher risk of developing cervical cancer in women living with HIV, in whom the cervical cancer mortality rate is two times higher. In the presented study CD4+cell count <200 cells/µl, persistent HR-HPV infection and HIV viremia>200 copies/ml, cigarrette smoking,  were independent and statistically significant risk factors associated with the development of CIN2+.

The analysis involved 599 patients after a long follow up and presents the important relationship between the immune status, the CD4+ cell count and the disease status.

It is interesting whether the patients were vaccinated against HPV?

And how the application of retroviral therapy after the new infection of HR-HPV might influence the outcome?

Author Response

1) It is interesting whether the patients were vaccinated against HPV? 

This condition was not considered in the analysis because the study was retrospective and it was based on  an historical cohort of women, enrolled in the pre-vaccination era.. HPV and even after the introduction of the vaccine, many WLWH did not undergo vaccination. 

2) how the application of retroviral therapy after the new infection of HR-HPV might influence the outcome?

In our analysis there are not women living with HIV who don’t take retroviral therapy for all study periods: for this reason we don’t compare the application of retroviral theraphy after the new HR HPV infection. Table 1 shows the therapy of women living with HIV at enrollment, but all women started the therapy after HIV diagnosis.

Reviewer 2 Report

The authors conducted a multicenter cohort study to evaluate the implications for the development of cervical precancerous lesions in HIV-infected women. A large cohort of women living with HIV (WLWH) with concurrent negative cytology and HR-HPV tests were followed over time to determine the incidence of CIN2 or higher histologically. The degree of smoking, HPV infection status and several clinical factors were examined. This study is very significant and is considered worthy of publication. As a reviewer, I would address a few questions.

1. In conclusion, the authors mentioned that "WLWH with good and stable immune status appear to have a low risk for high grade cervical lesions." After discussing high-risk factors, the authors describe low-risk factors only at the end, which may be somewhat confusing to understand.

2. The authors used CD4+ cell count <200 and HIV viral load >200 copies/ml as thresholds for HIV status. Prior publications which adopted this definition need to be described.

3. The results of the univariate analysis are presented in Table 2. Is it possible to perform a multivariate analysis after organizing the parameters?

4. In the number of PubMed searches, "ART HIV" is more common than "HAART HIV." I assumed that ART (Antiretroviral therapy) is used instead of HAART in recent years.

Author Response

1)In conclusion, the authors mentioned that "WLWH with good and stable immune status appear to have a low risk for high grade cervical lesions." After discussing high-risk factors, the authors describe low-risk factors only at the end, which may be somewhat confusing to understand.

  1. R) We modified the conclusion in order to a better understandig 

2) The authors used CD4+ cell count <200 and HIV viral load >200 copies/ml as thresholds for HIV status. Prior publications which adopted this definition need to be described.

The references were added in the text. 

3) The results of the univariate analysis are presented in Table 2. Is it possible to perform a multivariate analysis after organizing the parameters?

Table 3 reported the multinomials stepwise multivariate analysis about the risk factors for CIN progression in WHWL. To evaluate the independent effects of age, CD4+ cell count, HIV viral load, HR-HPV infection, and smoking, we used a stepwise ordered logistic regression model. CIN incidence was modeled as a 3-level variable (no CIN, low grade, and high grade CIN)

4) In the number of PubMed searches, "ART HIV" is more common than "HAART HIV." I assumed that ART (Antiretroviral therapy) is used instead of HAART in recent years. 

The use and type of antiretroviral therapy were recorded on the basis of the medications listed in the patients’ chart. HAART was defined as the co-administration of two nucleoside reverse-transcriptase inhibitors and at least one of the following: a protease inhibitor, a non-nucleoside reverse-transcriptase inhibitor, or an additional nucleoside reverse-transcriptase inhibitors. All other types of ART (mono- or dual-nucleoside reverse-transcriptase inhibitors) were defined as pre-HAART.